# Security Context Migration in MEC: Challenges and Use Cases

Wojciech Niewolski [1,2] , Tomasz W. Nowak [1,*] , Mariusz Sepczuk [1], Zbigniew Kotulski [1], Rafal Artych [2], Krzysztof Bocianiak [2] and Jean-Philippe Wary [3]

1 Faculty of Electronics and Information Technology, Warsaw University of Technology, 00-665 Warsaw, Poland
2 Orange Polska S.A., 02-326 Warsaw, Poland
3 Orange Innovation, 92320 Châtillon, France
* Correspondence: t.w.j.nowak@gmail.com

**Abstract:** Modern and future services require ultra-reliable mobile connections with high bandwidth parameters and proper security protection. It is possible to ensure such conditions by provisioning services in the Multi-Access Edge Computing system integrated with fifth-generation mobile networks. However, the main challenge in the mentioned architecture is providing a secure service migration with all related data and security requirements to another edge computing host area when the user changes its physical location. This article aims to present the state of research on the migration of the security context between service instances in Edge/MEC servers, specify steps of the migration procedure, and identify new security challenges inspired by use cases of 5G vertical industries. For this purpose, the analysis of the security context's structure and basic concept of the Security Service Level Agreement was performed and presented in the document. Next, a further investigation of the security context was conducted, including requirements for its reliable migration between edge serves instances. The study mainly focused on crucial migration challenges and possible solutions to resolve them. Finally, the authors presented how the proposed solution can be used to protect 5G vertical industries services based on several mobile use cases.

**Keywords:** 5G mobile communication; communication system security; MEC; mobile computing; security context; SLA; security context migration; vertical industries

## 1. Introduction

From the end user's point of view, mobile network services use resources in the cloud. The user is interested in the quality of the service guaranteed in the contract with the service provider and the service continuity of operation. Service providers use edge servers and Multi-access Edge Computing (MEC) technologies to ensure the quality of service [1]. User mobility requires service providers to follow the client with the service edge location, thus transferring user data, including security-related data, between edge servers. The support for user mobility has its roots in 2G roaming and handover procedures [2,3]. These services work daily with already deployed mobile networks with proven scalability and reliability. However, these solutions cannot be easily implemented in a generic MEC-based architecture.

The necessity to transfer the user data (sometimes called the user context) has already been observed in IP access networks supporting host mobility [4]. It was an alternative to reestablishing a service from scratch, requiring signaling flow for security services, which would slow the establishment of the mobile host. Examples of such services are Authentication, Authorization, Accounting (AAA), Header Compression, and Quality of Service (QoS). As a critical element of network security, the context transfer procedure in IP networks has become the subject of scientific studies on its security [5,6]. Moreover, some patents proposing algorithms and devices support the secure transfer of contextual data [7,8]. Several context transfer protocols were proposed, e.g., authorized [9] or privacy-preserving [10] procedures.

The goal of our study is to construct a context transfer procedure in terms of service security (i.e., the security context) between instances of edge servers in the 5G edge computing ecosystem [11]; see Figure 1. The first step is to determine the conditions of such a process that guarantees meeting the user's and service provider's requirements and the safety of the process itself. In the next step, such a procedure should be adapted to the specific needs of a particular use case of a network service, seen as implementing the indicated 5G virtual industry procedure. In this paper, we identify the main stages of such a process and present the challenge of its implementation in line with our expectations, suggesting them for seven different use cases.

The main novelty of this paper is the introduction of a security context relevant to applications hosted in 5G MEC and the problem statement of ensuring an agreed service security level when the migration of applications between MEC instances occurs. We stipulate that applications can be migrated between MEC instances, and respective security objectives can be achieved with different mechanisms available in different edge environments. To the best of our knowledge, there is very limited work on the security of migration of edge applications in the multi-cloud environment, as most of the existing solutions assume user context transfer due to devices' mobility between pre-configured application instances. While mobile networks provide continuity of data and voice services in case of device mobility, such seamless continuity should be delivered for the security context of migrating applications and managed in a decentralized manner. The described problem is positioned in the business environment where the Security Service Level Agreement formally expresses security requirements, and the proposed generic procedure for security context migration is applied to multiple Vertical use cases.

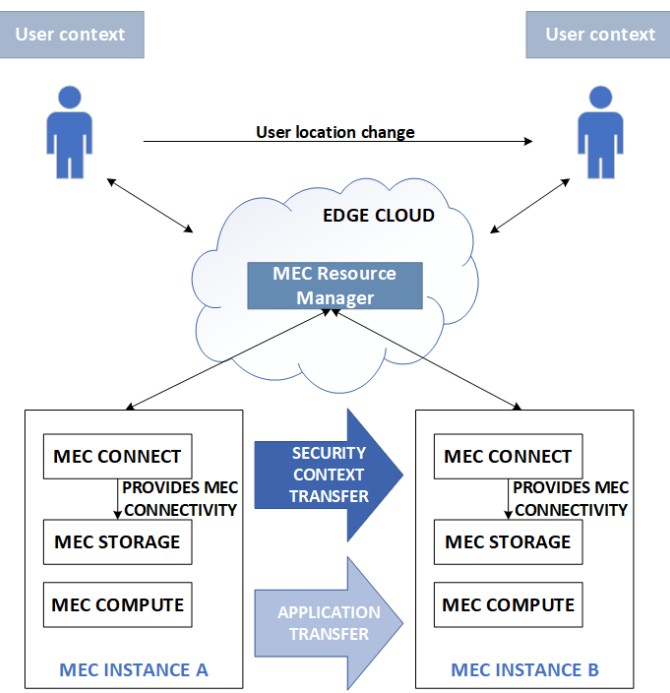

**Figure 1.** The scheme of security context transfer in the 5G edge computing ecosystem.

The rest of the paper is organized as follows. Section 2 presents the basic concepts enabling the construction of an ecosystem of mobile applications in the cloud computing environment equipped with edge servers such as application contextual data, security context, and SLA agreement. It also presents the main quality measures proposed by international standards, considered when determining the SLA to maintain the required level of security and the procedure for selecting the optimal set of such measures for a specific application. Section 3 presents the research on the context transfer related to services implemented in heterogeneous and mobile networks, standardization recommendations for

migration of the security context between different cloud service locations or edge servers, and related work of other authors on the migration of the security context. Section 4 contains the model's security context migration procedure. The proposed solution does not depend on the resource migration technology used and allows the construction of a migration method suitable for any location at the required level of security. Section 5 identifies six major areas of interest of context migration. Several challenges have been presented for each area that needs to be resolved to effectively and securely migrate the security context between edge instances. How the migration procedure in the proposed six domains of interest looks for specific applications is presented in Section 6. Several cases of specific mobile services of different vertical industries requiring the migration of the security context are considered, and the security challenges they face are presented. The next Section 7 proposes general solutions to the identified security challenges for the six context transfer areas of interest and discusses three selected use cases with their security solutions. The final Section 8 summarizes the paper and indicates the direction of further work.

## 2. Security Context and Service Level Agreement

### 2.1. Contextual Data in 5G MEC

As context information, we understand any information that can characterize an entity's state. An entity might be an asset of a computing system, such as a user, software, hardware, media storage, or data [12]. We can identify the following four basic categories for context information [13]:

- System context—applies to any information related to a computing system and a communication system;
- User context—refers to any context information related to the user and characterizing her/him;
- Environmental context—consists of any context information related to the physical environment, which is not covered by the system and the user context;
- Temporal context—defines any context information related to time.

Among the many categories of contextual data, a group likely to be migrated can be distinguished. The migration ensures a business continuity of services, which is particularly important in the 5G MEC network. Of course, the degree of such migration (e.g., the volume of data, type of data, etc.) depends on the type of MEC service and migration. Thus, the 5G MEC migration context data types are as follows:

- User equipment data describes information about the device that will be the client of the service.
- Subscriber data describe information about the user using the selected service.
- Subscription data describe information about the configuration of the service being provided.
- Quality of Service (QoS) data describe information about the required quality with which a service must be provided.
- MEC service-related data describes MEC infrastructure information related to the provided service.
- Spatial-temporal data describe the time and location information related to the use of the service.
- Mobile access restriction data describe access control information related to the provided service.
- Network service data describe network information related to the provided service.

In all categories listed earlier, it is possible to find data related to security and referred to as the security context.

## 2.2. Security Context, Its Components, and Migration

Each service requires an appropriate level of security. It usually depends on the data that this service will use. For example, medical data require more emphasis on confidentiality than crop data (data related to crop irrigation) [14]. From the point of view of the service provider, when concluding a contract for a given service, it defines the provided Service Level Agreement (SLA). On the other side, the service recipient may require a defined level of service quality contained in the contract. An analogous situation can be in the area of security. Security SLA (SSLA) defines the minimal protection level that must be provided with the service to secure it adequately. Of course, such a protection requirement can be achieved differently. For example, creating a secure communication channel service provider requires at least TLS 1.1. Thus, this requirement is fulfilled when TLS 1.1, TLS 1.2, or TLS 1.3 is used. With this in mind, we can define the security context as *a set of requirements that allows us to provide the appropriate security level for a service*. Generally, these requirements are met by the parameters (attributes) used to secure the service. The selection of parameters can be made using expert knowledge or formal models [15], allowing for the optimization of the number of parameters while ensuring maximum information on the security level.

Table 1 contains examples of main data related to the security context: security service, methods, and parameters [16–19].

**Table 1.** Security context examples.

| Security Service | Methods | Parameters |
| --- | --- | --- |
| Confidentiality | Encryption | Algorithm key length, type of encryption, type of algorithm |
| | Anonymization | Method of anonymization |
| Integrity | Hash function | Hash function hash length |
| Availability | Data/storage redundancy | Type of backups, number of backups |
| | Uptime | The total amount of time that the data are available for in end-use applications |
| | Usage RAID against data destruction | RAID level |
| Authentication | Identity verification and confirmation | Method of authentication and related parameters such as password, PIN, challenge-response, etc. |
| Access Control | Access verification | Method of access control (MAC, DAC, RBAC, ABAC) and related parameters such as roles, attributes, data classifications (e.g., public, private, secret, top secret), etc. |
| Non-repudiation | Digital signature | Algorithm key length, type of algorithm |
| Isolation | Trusted Execution Environment (TEE) | Type of hardware technology, encryption key required for data protection |
| Traffic analysis | Data leak protection | Security policy in a DLP solution related to characteristics data |
| | Gathering logs and events | Security policy in SIEM solution related to logs/events analysis and correlations. |
| | Data filtration | Security policy in DBF solution related to data protection against leakage |

In this paper, we will understand the *security context migration* (also called security context transfer) as a *procedure that moves the security context to a specified target environment and implements the security context in the target environment*. The set of requirements that constitutes the security context might have various implementations in different environments

and deployment locations, so the aim of migration is to setup the proper implementation of those requirements in the target's place.

*2.3. Service Level Agreement*

A Service Level Agreement (SLA) is an agreement between a Service Provider and a Service Recipient regarding the service's quality level. It is essential in mobile services and cloud-based solutions, where the service context dynamically changes or is hard to establish. According to ISO/IEC standard [20] concerning clouds, the SLA is a part of the Cloud Service Agreement that includes Service Level Objectives (SLO) and Service Qualitative Objectives (SQO) for the covered services. The SLO is a commitment the service provider makes for a specific, quantitative characterization of the service, where the value follows the interval or ratio scale. The SQO is an analogous commitment for a specific, qualitative characteristic of a service, where the value follows the nominal or ordinal scale. The SLO can be expressed as a range, while SQO is an enumerated list. A corresponding NIST document [21] presents a business approach to the services assuming that they are determined by a legally binding agreement between the two parties, called a Service Agreement. The agreement includes the SLA as the technical performance promises made by a provider and remedies for performance failures. The promises made explicitly to consumers include availability, remedies for failure to perform, data preservation, and legal care of consumer information. The SLA can also contain promises explicitly not made to consumers (limitations), including scheduled outages, force majeure events, service agreement changes, security, and service API changes. Concerning this document, we can see that security issues are included, to some extent, in each class of rules mentioned above. Both parties, the consumer and the service provider, are responsible for the security level in SLA.

To define specific elements in SLA (the metrics), we must perform a systematic procedure in order to note critical information and to reduce redundant information. For instance, this can be performed according to the procedure proposed in [22]. Thus, choosing SLA Metrics can run in three steps; see, e.g., [22].:

1. Group metrics into broad categories;
2. Select the most appropriate metric(s) from each category;
3. Combine them into a "balanced scorecard" for the project.

In practice, we implement the procedure that leads to the optimal security measure choice that is non-overlapping and covers the entire domain of interest. This can be performed in four steps:

1. **Select the categories of metrics**

   - Define several categories of metrics covering the main fields of interest of the agreement.
   - Be sure they cover the entire criteria space and do not overlap.

2. **Selection of metrics in each category**

   - In each category, choose the best metrics to measure the criteria.
   - Limit the number of measures.
   - Be sure they do not overlap.

3. **Select the methods to express each metric**

   - Select the measures of SLO and SQO types.
   - Use SLO metrics for which the values are unique and the measurement results cannot be questioned, and use SQO otherwise.

4. **Check if the metrics satisfy expected quality conditions**

   - Accuracy of measurement;
   - Expected precision;
   - Objectivity of presentation;
   - Durability of properties, etc.

The crucial step in this process is a choice of the SLA metrics categories. In Ref. [22], the following four necessary categories have been proposed.

**1. The volume of work.** The purpose is to reflect the parties' required activity level. If the activity consistently exceeds this level, the parties should renegotiate the existing SLA. The metrics must refer to the number of units of a work product or the deliverables per unit of time. They include the number of support calls per month or the number of maintenance requests per unit of time.

**2. Quality of work.** The purpose is to cover a wide range of work products, deliverables, and requirements. Each deliverable should have quality acceptance criteria. Quality metrics measure the provider's conformance to a standard. They refer to defect rates, standards compliance, technical quality, service availability, and customer satisfaction. The metrics express quality positively (percentage of deliverables accepted) or negatively (percentage of deliverables rejected).

**3. Responsiveness.** Metrics in this category measure the time it takes to complete a task or satisfy a request. They include time-to-market, time-to-implement something, time-to-acknowledgment, and size of the backlog. Such metrics figure prominently in consumers' and clients' perceptions of the quality of service delivered. Responsiveness is often a key reason why companies choose to outsource work initially.

**4. Efficiency.** Metrics in this category measure a provider's effectiveness at delivering services at a reasonable cost. Key metrics include cost/effort efficiency (cost per support call) and team utilization (percentage of time spent on support).

*2.4. Security Service Level Agreement*

ISO/IEC standards [20,23] propose a series of Service Level Objectives (SLO) and Service Qualitative Objective (SQO) for use in SLA. In practice, the SLA applies to a specific network service or a specific area of quality of service. In this subsection, we present a set of these quality measures used to assess the level of security, i.e., potentially constituting elements of the constructed SLA for the guaranteed level of protection. Security meets the basic requirements of confidentiality, integrity, and availability of information (CIA) and ensures access control to resources or protecting user privacy—Personally Identifiable Information (PII) protection. We will call such an SLA the Security Service Level Agreement (SSLA in short). In some papers (see, e.g., [24]), the authors consider the SSLA as a process where the application's security levels, controls, and metrics are specified at the SSLA creation process and continuously monitored at runtime once application components are deployed over the cloud ecosystem. In this paper, we consider the SSLA as a set of constraints and not a process of their monitoring.

Essential SSLA components of SLO type are presented in Table 2. These include metrics characterizing data availability, requiring appropriate adjustments to the service, and describing the level of protection of data processed in the service and the user's private data. A relatively new protection criterion is the degree of isolation [25], which is particularly important in modern mobile networks [26]. It is also essential to protect privacy, which is regulated by the legislation of many countries and industry organizations' standards. Except for metrics of the crucial components, we can consider on-demand SSLA components of the SLO type. They can be the Key Performance Indicators (KPIs) for security operations and incident response presented in Table 3 and the maintenance measures representing the security program's effectiveness, contained in Table 4. These measures are suitable for mitigating overall risks.

**Table 2.** SSLA basic components (SLO type).

| Component (Origin) | Metrics | Description |
|---|---|---|
| Availability | Availability, all aspects | Dependent on service provider, external services, environment, user equipment, malicious activities |
| | Abandonment Rate | Percentage of calls abandoned while waiting to be answered |
| | ASA (Average Speed to Answer) | Average time (usually in seconds) it takes for a call to be answered by the service desk |
| | TSF (Time Service Factor) | Percentage of calls answered within a definite time-frame |
| | FCR (First-Call Resolution) | A metric that measures a contact center's ability for its agents to resolve a customer's inquiry or problem on the first call or contact |
| | TAT (Turn-Around Time) | Time taken to complete a certain task |
| | TRT (Total Resolution Time) | Total time taken to complete a certain task |
| | MTTR (Mean Time To Recover) | Time taken to recover after an outage of service |
| Non-disclosing consumer data | Integrity | Hash function of certain hash length |
| | Confidentiality | Block or stream cipher of certain secret key length |
| | Isolation | Security isolation, performance isolation, error propagation isolation |
| Personally Identifiable Information (PII) | PII, general | Under GDPR and domestic legal regulations in details |
| | Geolocation | Expected accuracy of determining user's device position in a fixed coordinate system |
| | Level of anonymity | Expected probability that an entity will be identified in a group of other entities |
| | Transparency | Describes the notification of PII collection and processing of PII |
| | Data Breach Notification Period | Description of the maximum length of time taken for the CSP to notify the CSC of the occurrence of a data breach involving PII |
| | Retention Period | Description of the retention limit of PII concerning the covered services |
| | Use Period | Description of the use limitation of PII concerning the covered services |
| | Disclosure Period | Description of the disclosure limitation of PII concerning the covered services |
| | Erasure Period | Description of the erasure limitation of PII concerning the covered services |

**Table 3.** On-demand SSLA components (SLO type); key performance Indicators (KPIs) for security operations and incident response [27].

| Metrics | Description |
|---|---|
| Mean Time to Detect (MTTD) | Average time to notice abnormal behavior indicating malicious, suspicious, or risky behavior. |
| Alarm Time to Triage (TTT) | Time to prioritize the alerts that indicate the highest risk. It indicates the high, medium, and low-risk alerts. |
| Alarm Time to Qualify (TTQ) | Time the security operations team needs to determine that an alert qualifies to be moved to the incident response team. |
| Mean Time to Acknowledge (MTTA) | Average time it takes the security operations and incident response team to acknowledge an alert before they begin doing an investigation. |
| Mean Time to Investigate (MTTI) | Average time it takes the incident response team to investigate an alert after it is acknowledged. |
| Mean Time to Resolve (MTTR) | Time it takes the incident response team to get from the investigation to the recovery step. |

**Table 3.** *Cont.*

| Metrics | Description |
|---|---|
| Mean Time to Contain (MTTC) | Time it takes the security team to locate the threat actors and prevent them from moving further into the systems and networks. |
| Mean Time to Recover (MTTR) | Time to complete the security and incident response processes and recover the affected system back to its pre-incident state. This metric functionally incorporates the MTTD, MTTI, MTTR, and MTTC. |
| Cost per incident | The amount of downtime, resources, and other activities associated with security incidents when calculating this amount. |
| Number of incidents per device or host | Visibility into how well the organization is monitoring and mitigating risks. Identifies devices or hosts that are more likely to experience an incident. |
| Mean Time Between Failures (MTBF) | Average time between system outage or repairable failures. |

**Table 4.** On-demand SSLA components (SLO type and maintenance measures).

| Component (Origin) | Metrics | Description |
|---|---|---|
| Data maintenance | Backup interval | Time between backup procedures of applications pattern and users' data |
| | Retention period for Backup | Data Minimal time of retention of backup sets |
| | Number of Backup Generators | A parameter which describes the performance of the backup process |
| | Backup Restoration Testing | The frequency of testing the operation of contingency plans |
| | Recovery Point Objective | A parameter which describes the depth of recovery process |
| | Temporary File Erasure Period | Time after which a non-used temporal file is securely destroyed |
| | Data Retention period | Minimal time when data not in use (end of the agreement, termination, etc.) can be restored from the system |
| | Log Retention Period | Minimal time when the log files can be obtained from the system |
| System maintenance | Elasticity | A property where resources can be rapidly and elastically adjusted to quickly increase or decrease |
| | Elasticity speed | Describes how fast a cloud service can react to a resource request |
| | Elasticity precision | Describes how precise the resource allocation meets the actual resource requirements at a given point in time |

In addition to the SLO measures of the security level, the SSLA may contain SQO measures used when it is difficult to quantify the protection level for the selected factor precisely. The list of such measures is included in Table 5. A checklist usually marks the fulfillment of the criteria recommended in it.

**Table 5.** SSLA components (SQO type).

| Component (Origin) | Factors | Description |
|---|---|---|
| Authentication | Authentication mechanisms | Selected from some portfolio |
| | Third party authentication support | KDC, PKI, OAuth 2.0, etc. |
| | Strong authentication | Multi-factor, Identity-based, etc. |
| On-demand security features | Evidence of Security Environment | Enabling the user to be informed whether a security feature is in operation or not |

**Table 5.** *Cont.*

| Component (Origin) | Factors | Description |
| --- | --- | --- |
| Personally Identifiable Information (PII) | Data Breach Notification Method | Confirmation, that the regulations of SLA related to PII are satisfied in the MEC instance |
| | Disposal Policy | A way to securely dispose of PII |
| | User present location | Availability of information about the user's current location |
| User data transferred | Application data | User data needed for present usage of an application |
| | Historical application data | User data used by applications in the past or backup information |
| Action confirmation data transferred | System logs related to the application | Confirmation if the data allowing legal procedures or cross-verification of security information is transferred to the MEC instance |
| | History of user locations | Availability of information about the user's previous locations |
| | History of inter-MEC transfers | Availability of information about the previous data transfers between MECs |

## 3. Related Work on Context Migration

The problem of context transfer has become particularly important now in the era of the development of fifth-generation and higher mobile networks [28]. It creates a need for safe, user-friendly solutions such as Shared Automated Mobility On-Demand (SAMOD) [29]. The development work in this area includes researching the Internet of Things and using edge servers, especially MEC technology. Paper [30] gives an overview of applications, architecture, advantages, and challenges in IoT networks, including context-related problems. The authors of [31] propose a four-layer framework that incorporates Software-Defined Networking (SDN) and Network Function Virtualization (NFV) to utilize their flexibility in making rapid adjustments to network conditions to support context-aware security in IoT applications. In [32], the authors survey standards, with particular emphasis on 5G and the virtualization of network functions and address the flexibility of MEC smart resource deployment and its migration capabilities. It can help optimize the cost of resources because MEC provisioning has to be carefully designed and evaluated. In [33], the authors propose MEC-based intelligent service migration architecture to improve the service continuity in multi-domain Long Term Evolution) and 5G cellular networks. Paper [34] presents a trust layer for public MEC infrastructure that handles establishing and updating trust relations among all MEC entities, making the interaction within an MEC network transparent. In [35], the authors focus on a trust mechanism based on interactions between MECs to increase reliability in the context of a service migration scenario. In [36], a context-aware distributed online learning algorithm for efficient content caching is proposed according to a novel tree-based and contextual multi-arm bandit theory for collaborative MEC. The authors of [37] present a resource allocation algorithm based on deep learning. The algorithm assigns user requests to the optimal server and allocates the optimal amount of resources to the user equipment based on a utility function. Paper [38] offers a decentralized authentication architecture that supports flexible and low-cost local authentication with the awareness of context information of network elements such as user equipment and virtual network functions. In [39], the authors propose a mathematical model for latency-optimal on-path allocation of VNF chains on physical servers within an edge network infrastructure, with special considerations for network security applications and operator's best practices. Security policies are the topic of paper [40]. This paper considers both the adaptive and cost-benefit aspects of security and introduces a context-aware technique for designing and implementing adaptive, optimized security policies. Paper [41] considers optimal planning for the deployment of the base stations by taking into account the mobility management of the users and the service degradation that this mobility process could cause. It proposes a Link-Network Assignment algorithm to optimize the assignment of the base stations to the access routers in the mobile network to reduce

signaling costs and packet delivery costs, which can help in decisions with respect to the user context's transfer initiation and destination.

Paper [42] describes a three-way decision-based service migration strategy in MEC. The concept of the paper is to introduce a delayed decision that does not ultimately decide if the service (context in our situation) should be migrated or not. This approach aims to reduce energy consumption and the time needed for migration. As an optimization goal, the energy consumed during the migration process is also discussed in, e.g., [43,44]. Another aspect that should be considered during migration is the proper placement of the MEC application according to its security constraints [45]. The authors of this paper consider different orchestration rules for various applications to fulfill the required parameters of the service and simultaneously guarantee the necessary security level.

In paper [46], the authors propose a five-step service migration procedure using the containerization of the software, distinguishing real-time and non-real-time code, and deploying it in an edge device and a cloud environment. The application of migration of entire virtualized services (including the context) has been proposed in several papers, e.g., in [47,48] with containers and [49,50] with virtual machines.

The development of mobile technologies on the web has made it necessary to propose standardized procedures for moving applications together with their resources, i.e., the user's context, between different locations, such as cloud computing or edge servers. The existing standards create a general framework for such a process, indicating the parties initiating and supervising the transfer and the available schemes of such a procedure. The ETSI standard [51] proposes three high-level implementation approaches for user context transfer where the MEC system is the decision maker and selects the appropriate MEC application instance. The device application-assisted user context transfer assumes that the application client is assisted by the device application associated with the MEC application in the MEC system.

The device application can receive the up-to-date information of the MEC application address and may pass this information to client-side applications. A client application designed to be assisted by the device application does not require the underlying access network and MEC to maintain the IP address of the application. In addition, the client application may use the new MEC application instance address for the user context synchronization in the new user application instance.

The MEC-assisted user context transfer relies on the Application Mobility Service (AMS) of MEC to trigger the user context transfer and to inform the MEC application of the target end point of the user context. The MEC application is a consumer of the AMS. The AMS is kept updated on the devices served by the MEC application. The AMS notifies the MEC application of the user context target endpoint when there is a need for a user context transfer. The MEC application then sends the user context to the target endpoint. The user context is application-specific and is exchanged between MEC application peers in the source and target MEC hosts.

The application's self-controlled user context transfer assumes the application (server side, client side, and centralized cloud component) can detect the need for the user context transfer by its means and execute the context transfer without assistance from the MEC system. The role of the MEC system is to fulfill the applicable service and session continuity commitments for the application traffic and to enable the required application communication.

The research coordinated by ETSI devotes much attention to the problem of secure migration in the mobile environment as one of the most critical stages of the edge application's lifetime [52]. Several ETSI standards consider different aspects of migration. The standard [53] formulates general requirements, specifies procedures, and presents several use cases for Inter-MEC systems and MEC-Cloud systems coordination. It proposes addressing general provisions of the MEC's reference architecture [54]. A MEC platform should be able to discover other MEC platforms belonging to different MEC systems and exchange information securely with other MEC platforms and other MEC applications. It

is assumed that MEC platforms in different MEC systems can discover each other without the involvement of the MEC system-level functional elements. The standard proposes the hierarchical inter-MEC system discovery and communication framework with a MEC system level inter-system discovery and communication and a MEC host level inter-system communication between the MEC platforms. The standard proposes several recommendations on how this goal can be achieved. The ETSI White Paper [55] attempts to standardize the context migration procedure indicating specific interfaces of the Synergized Mobile Edge Cloud architecture to assist in context migration but finds the entire subject of context migration complex. The widespread implementation of the MEC federation concept [56] may, in the future, make it much easier to migrate applications and the associated security context between MEC instances managed by different operators.

The ETSI specification [57] focuses on the V2X (Vehicle-To-everything) communication and scenarios. It considers cases of Vehicle OEM or ITS (Intelligent Transport System) operator as a VIS (V2X Information Service) provider with a single or multiple MNO (Mobile Network Operator). The VIS service aims to reduce latency by better handling the MEC environment flow and roaming scenarios. The specification assumes that the MEC VIS element could be a part of the discovery framework for services.

Security context migration is also essential to network security practitioners and cloud service providers. It is due to the need to facilitate the security of the provided services and make the cloud offers more attractive for end users. In the presented practical solutions (see, e.g., [58–61]), compliance with standards and SLA requirements is not a priority. It is crucial to efficiently and safely use the specific cloud environment offered by the provider. In this paper, we propose the security context migration procedure, which is compatible with standards and is under the control of the SLA constraints. Examples of implementing this procedure dedicated to specific verticals and the relevant safety requirements should lead to models for which validation and the proof of security are possible.

## 4. Procedure of Security Context Migration

Based on the recommendations presented in the standards for user context migration and the determination of the requirements for ensuring the guaranteed level of service (SLA), we can propose a procedure for migrating the security context between instances of edge servers. This procedure aims to maintain service continuity and guarantee the required level of security as defined in the SSLA during security context transferring and regular service operation.

The security context migration procedure can be considered a management cycle with seven steps; see Figure 2.

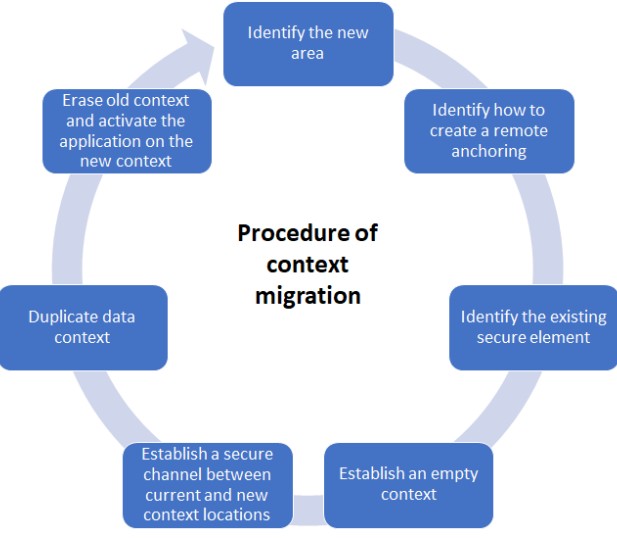

**Figure 2.** The life-cycle of the security context migration.

1.  **Identify the new area**
    - Decide the destination of the application instance transition, e.g., from the following:
        – One access point to another;
        – One MEC instance to another, etc.
    - Identify the resources of the new destination.

2.  **Identify how to create remote anchoring**
    - Decide which level of the network will be used to transfer the security context:
        – User application self-controlled;
        – MEC Application controlled/MEC system controlled;
        – MEC platform controlled;
        – MEC host-level controlled.
    - Compare the resources of the present location and the new destination.

3.  **Identify the existing secure element (possibly performed together with Step 2)**
    - Check the SSLA constraints.
    - Fix which resources constitute the security context.
    - Extract parameters and resources of the security context for the assumed layer of the security context transition.

4.  **Establish an empty context**
    - Fix which parameters and resources of the security context are required in the new application instance's location.
    - Initiate the new configuration with local parameters and data repositories and open it to transfer the security context data.

5.  **Establish a secure channel between current and new context locations**
    - Establish how to deliver the security context data to the new application instance location:
        – Construct a secure communication channel;
        – Create access to a common data repository, PKI, KDC, etc.
    - Check the correctness of the channel.

6.  **Duplicate contextual data**
    - Obtain keys or security credentials from PKI, KDC, or repositories.
    - Install the delivered parameters, if applicable.
    - Initiate alternative solutions if needed.

7.  **Erase old context and activate the application on the new context**
    - Check the new SSLA and decide if the requirements are satisfied.
    - Start the application in a new security context instance location.
    - Securely close the old instance of the application's location.
    - Check if the procedure is correctly closed.
    - Send PII message to the stakeholder if it is required (legal regulations).

The described steps could be extended with additional algorithms, e.g., decision algorithms that validate whether the security context's migration should be performed in current circumstances. For the decision-making process, the following might be considered: the energy consumption expected lifetime of the context in the target place, current load in the system, contractual obligations, etc.

## 5. Areas of Interest of Security Context Migration

The security context's migration process is not only about data transfer. The entire operation should be prepared before such a transfer and carried out end-to-end. We

organize the preparation areas and stages of the context transfer implementation into six main areas of interest. These are the following:

- SSLA covering conditions that should not deteriorate;
- Contents to transfer, including indispensable services and data;
- Transfer initiation to make the optimal decision;
- Transfer process including all its stages;
- Transfer management to make the process end-to-end;
- Isolation of the process to guarantee its safety.

Below, we discuss the steps of preparing and migrating a security context. We present the challenges that must be addressed to ensure service continuity, for which its security context requires migration between edge servers. In the following sections, we will specify these challenges for selected use cases of mobile services and propose some solutions.

### 5.1. SSLA

What should happen if the migrated security context does not meet SSLA?

- Whether service provision should be automatically terminated, or a limited-service function may be allowed to be used until the security context does not meet the SSLA (after another migration or context update)?

How to ensure at least the same level of data protection during and after migration?

- How to ensure security context security at rest and in transit?

How must the SLA be formulated to satisfy PII management conditions?

- Information for the end-user and where his PII is located now.
- Conditions to guarantee removing not-used PII.

What are the parties of SSLA?

- This at least contains End-user, service provider, and MEC infrastructure provider. What other stakeholders (parties) should be included?
- How to construct SLA for three or more parties?

How to prioritize SSLA conditions to enable service to the maximal extent?

- If there are no agreements between network operators or MEC providers?
- If available resources are not equal/equivalent/comparable?

Which data are required to prove satisfying SSLA?

- How to provide its non-repudiation?

How can the providers guarantee/verify satisfying SSLA requirements at the users' discretion?

### 5.2. Contents to Transfer

This category is the central point of the transferring process. It contains the following:

- PII data that are very sensitive due to legal regulations;
- User data, both private and related to the usage of the application;
- Application data required for the application's work;
- Service provider's data required to provide service continuity and to satisfy the SLA;
- Security credential providing service's security;,
- Data enabling a proof that SLA has been satisfied.

Relating to the migration of contents, we have the following:

- What type of context information should be migrated (firewall rules, virtual machines, Docker images, etc.)?
- How to deal with conflicts, e.g., full VLANs, that are already used network ranges and addresses?
- How to detect relevant security context related to a given UE and service?

- Which critical security parameters must migrate and which must be newly created/taken from a repository/PKI/KDC?
- How to share security context data between a new instance and some archive repository?
- Is inter-MEC data sharing or common data repositories resonable?
- When do we delete the security context from the previous MEC instance?
- How to build a structure/prioritize security context data to guarantee service continuity during the transfer?
- How to share the security context data between the parties (user device/MEC hosted service/MEC platform)?

*5.3. Transfer Initiation*

Before starting the data migration, there must be some factors that will initiate this process. Thus, the challenge here refers to the following aspects:

What conditions must be met to migrate the security context:

- Should the UE or the MEC trigger migration?
- What checks must be performed on UE and MEC sides to start the migration?

Decide when to begin transferring security context and who/what is responsible for it:

- Before or after connecting the UE to a new MEC instance?
- Should migration start as soon as the UE joins the new MEC instance?
- Should the migration start earlier, for example, based on some prediction of selecting a new MEC instance?
- When to start the transfer? What factors should decide about it (e.g., user location, MEC present workload, expected future location, business policy, expected future time of the session, and expected next links)?
- What triggers the evaluation process that results in the decision that the context has to be migrated?
- Who executes the evaluation process?

*5.4. Transfer Process*

What should happen when a security context migration is interrupted:

- Should the migration be repeated or UE requested about the current security context?
- How long does the old MEC instance have to store the security context?
- Whether the re-migration should apply to the entire security context or only selected data?

Migration time-related questions:

- When to run a migration?
- How to migrate a service within a time required by service (seconds and milliseconds)?
- How to make the data available in the new location on time?

*5.5. Transfer Management*

Decide which instance is responsible and controls the security context transition:

- User device application:
  - What is the maximum expected security level in such a case?
- MEC-hosted application:
  - Who is responsible for SLA in such a case?
- MEC platform:
  - Who is responsible for SLA in such a case?
  - How to construct the MEC federation to enable security context transfer?
  - What are the details of building hierarchical security context transfer?
  - What is send at each level (user application, MEC application, and MEC platform)?
  - What if the user application must be changed?

- What if the MEC application must be changed?
- What if a common MEC federation cannot be created?

Where to migrate the context?

- How to select the best MEC Server? What about the hysteresis loop?
- What if UE lost the connection with MEC Server?
- Is it possible to migrate the context to the cloud instead of the MEC Server?

Selection of possible targets:

- How to determine that target MEC Server satisfies SSLA of migrated service?
- How to deal with differences in implementation—on hardware and software levels?

### 5.6. Isolation

Another challenge is related to data isolation. MEC instances can store many data, so it is important to isolate and adequately identify them. The constraints on isolation are as follows:

- How to transfer the security context to guarantee the required isolation level?
- How to measure the level of security context isolation after the migration is complete?
- How to ensure isolation during transfer on-the-fly?

## 6. Vertical Industries Use Cases and Security Context Migration

The purpose of this section is to identify the most critical challenge in solutions in the process of migrating the security context between MEC instances. We formulate these challenges at an intermediate level of detail, that is, for the six areas of interest of the security context migration defined in Section 5. To better identify the challenges, we consider them for different use cases in several 5G MEC virtual industries and then validate their significance to make it easier to propose possible remedies.

### 6.1. 5G MEC Verticals and Use Cases

The business value of a computer or mobile network is the ability to provide services for clients and customers. Significant groups of clients are called 5G vertical industries or verticals [62]. The following example set of verticals covering the entire spectrum of modern IT was proposed in the paper [63]:

- Manufacturing Industry;
- The Financial Sector;
- Healthcare;
- Retail;
- Telecommunications;
- Authorities;
- Media and Entertainment;
- Smart City;
- Agriculture and Food Industry;
- Logistics;
- Education, Culture, and Science;
- Critical Infrastructure Sectors.

The use cases considered in this section represent situations where mobility and using MEC service are crucial for the effectiveness of the communication purpose of the networked application. Table 6 describes the MEC to mobile characteristics, communication requirements, and contextual data that could be considered for migration for the following use cases.



**Table 6.** Characteristics and requirements for MEC communication and data considered for migration.

| Use Case | MEC to Mobile Characteristics and Communication Requirements | Contextual Data Considered for Safe Migration |
|---|---|---|
| Mobile-to-Bank (M2B) | - MEC services can obtain security context from user devices via POSes and accept or decline further transactions.<br>- After transaction requests from the user's device, via POS or direct link to user's account, provide the security context via MEC-to-MEC inter-banking communication. | - User-device-supported security solution and payment service security credentials carried by the device.<br>- Security context parameters delivered by a bank through the inter-MEC network are used to establish a secure link with the user's device and check its security credentials. |
| Remote monitoring of health or wellness data through wireless devices | After the detection of a patient state parameter change, the security context is transferred to the proper MEC application from user devices. | - Patient's health data condition.<br>- Secure context that refers to the following:<br><br>• Creation of a secure communication channel (data includes encryption, authentication, and non-repudiation parameters)<br>• Proper isolation level<br>• Data monitoring requirements (security policy parameters for data access control software or hardware and data returned in response to a request) |
| Wireless telesurgery | - Ensuring adequate data quality may require communication between other applications within the given MEC Host.<br>- Ensuring adequate data quality may require communication between other applications within the given MEC Host | - Location and time data<br>- QoS transmission parameters<br>- Security context that refers to:<br><br>• Creation of a secure communication channel (data includes encryption, authentication, and non-repudiation parameters) for the transmission of audiovisual data and robot control commands<br>• Data monitoring requirements (security policy parameters for data access control software/hardware and data returned in response to a request)<br>• Parameters related to the high availability of a service |
| Critical communication (emergency) | MEC host must be able to manage the network locally, restore the signaling common network and provide security to mobile devices | Security context migration at the MEC host level, possible recovery data for the entire network, renewing/restoring security context information, initiation of new, used devices |
| City surveillance | - The mobile device can switch between MEC cells that requires proper context handover, data buffering in case of lack of connection, and data exchange between MEC cells, to correlate information obtained by the device<br>- The data might be streamed to patrols to monitor the situation remotely | - Current location, angle, velocity, acceleration<br>- Task category (confidential, secret, incident management)<br>- Current information context sent to previous MEC servers |
| V2X communication | - Ultra-low latency for communication with MEC<br>- High integrity of the communication (TTP)<br>- Fast context handover that will not affect the communication V2X<br>- Consciousness about assignments to MEC surrounding devices. | - Current location, angle, velocity, acceleration,<br>- Battery level<br>- Awareness of surrounding devices and objects<br>- Knowledge about trust level related to other vehicles, objects |

**Table 6.** *Cont.*

| Use Case | MEC to Mobile Characteristics and Communication Requirements | Contextual Data Considered for Safe Migration |
|---|---|---|
| Automated vehicles | - Ultra-low latency for communication with MEC<br>- High integrity of the communication (TTP)<br>- Fast context handover that will not affect the communication between vehicles, infrastructure, and pedestrians' devices.<br>- Consciousness about assignments to MEC surrounding devices and vehicles.<br>- Ability to negotiate the cooperative action's protocol—many manufacturers on the market. | - Current location, angle, velocity, acceleration<br>- Battery level<br>- Awareness of surrounding devices and objects<br>- Knowledge about trust level related to other vehicles, objects |

*6.2. Mobile-to-Bank (M2B)*

Banking services are widely available today on mobile devices that can be used to perform financial transactions or access a bank account. It also could be managed from regular workstations with a web browser or with payment technologies such as credit cards in Points of Sale [64]. Low-cost payments (below some limit and below some number of transactions in a period) can be made autonomously, and the security context can be transferred on the user-device level. Other transactions (above limits) need contact with the bank's infrastructure, common bank infrastructure can be used, and security context transmissions are performed on the MEC platform level using the dedicated inter-bank signaling infrastructure. The required connection quality parameters are not so high compared to other banking use cases [63], but MEC technology could still support this use case because it is an example where the eMBB technology might be applied. Services hosted in the MEC can improve the QoS of performed transactions or enhance security by exposing security services to low-computing devices.

In the BFSI, all services and even single operations are performed according to legally justified agreements, so some challenges are solved "by definition". On the other hand, the core BFSI network must be strongly isolated even if it needs access from untrusted user equipment, so isolation is critical for security (and SSLA).

In Table 7, there are security challenges for the use case presented for every area of the security context transfer considered in Section 5.

**Table 7.** Challenges in the Mobile-to-Bank (M2B) use case.

| Challenge | Challenge Impact on the Use Case | Severity |
|---|---|---|
| SSLA | The conditions of SSLA must be included in all services agreements, precisely defining the responsibilities of the parties. Eventually, risk sharing is used for insurance of the processes. | L |
| Contents to transfer | Data required in BFSI services are defined by legal regulations and internal policies | L |
| Transfer initiation | The service would be probably performed by BFSI-dedicated MEC hosts with a limited number of instances, so the decision procedure would be easier than in open networks | M |
| Transfer process | Depending on what is the level of management (User application/MEC platform) | H/M |
| Transfer management | Depending on what is the level of management (User application/MEC platform) | H/M |
| Isolation | Providing isolation is crucial for security, especially on the line User Device—MEC and during switching Uds from MEC to MEC. | H |

### 6.3. Remote Monitoring of Health or Wellness Data through Wireless Devices

Wireless IoT devices can be used to monitor a patient's health status [65,66], collect datasets, analyze them, and execute actions with or without a doctor's assistance [67,68]. In emergency situations, the reaction time is very important and significantly affects the prevention of health damages and recovery times [69]. It requires continuous access to IoT devices with patient sensors and high service availability. Due to the fragility of the data, proper data management and security are needed [70].

In the remote monitoring of health or wellness through wireless devices, challenges such as defining SSLA, choosing contents to transfer, transferring process, and ensuring proper isolation levels are very critical (the level of severity is high). This is due to the high level of security that health data require. Moreover, the service itself is crucial, as it may directly impact the patient's health condition; therefore, the transfer procedure must be reliable and properly carried out (with all necessary data) [71].

In Table 8, there are security challenges for the use case presented for every area of the security context transfer considered in Section 5.

**Table 8.** Challenges in the remote monitoring of health or wellness data via wireless devices use cases.

| Challenge | Challenge Impact on the Use Case | Severity |
| --- | --- | --- |
| SSLA | Data on health are sensitive data and should be adequately protected. It requires appropriate SSLA requirements. Strong security methods must be used to protect these data, which can sometimes be challenging for sensors. | H |
| Contents to transfer | The selection of data for remote monitoring of the patient's health is essential in the context of its analysis and selection of treatment. Data that are too small may result in an incorrect treatment choice and, consequently, may harm the patient. In addition, the security of these types of data must meet the relevant security requirements, e.g., those contained in the HIPPA standard. Therefore, a security context must be sent in addition to the patient's data. It will be possible to determine whether the patient's data can be protected at a sufficiently high level. | H |
| Transfer initiation | Before the context transfer is started, the side that will initiate it must be selected: sensors or a device collecting data from sensors measuring the patient's condition or a MEC instance. It is related to specifying the requirements when such a transfer could occur. On the one hand, it may happen cyclically, i.e., data will be collected every specific time or whenever the sensor detects a change in the patient's condition, which may negatively impact him/her. | M |
| Transfer process | The patient's data must be correctly migrated during the transfer process. It is essential because we can react quickly to rapid health changes. In cases where specific data are missing, an attempt to obtain them should be made immediately. | H |
| Transfer management | Managing the transfer of patient data is important for the proper operation of the service. Therefore, choosing which party is responsible for this process must be made. One of the challenges in this process will be the selection of the appropriate MEC instance where the data will be stored, and this instance will meet the requirements specified in the SSLA. | M |
| Isolation | The entire process of collecting patient data is associated with their appropriate separation. Other patients' data should not be mixed with our data, and such a set will be analyzed. Therefore, the isolation of patient data during its transfer and its storing and processing is required. | H |
| Isolation | Providing isolation is crucial for security, especially on the line user device—MEC and during switching Uds from MEC to MEC. | H |

### 6.4. Wireless Telesurgery

This use case covers audio–video live stream between the patient, on-site medical team, and remote medical team members, particularly doctors [72]. It might be possible to use robots to perform the operation from the remote location [73,74]. Another scenario is to conduct an instant meeting between the ambulance's crew and the doctor to receive fast advice on how to proceed with the patient. In both scenarios, ultra-low latency and high availability are crucial to providing such services [75].

In wireless telesurgery challenges, all challenges have a high severity. If we were to consider only the category related to the transmission of sound and image, the severity would be minor. Still, the telesurgery service is so critical that the entire process of data migration and related operations must be reliable. Therefore, migration should be secured appropriately to meet high availability and reliability requirements.

In Table 9, there are security challenges for the use case presented for every area of the security context transfer considered in Section 5.

**Table 9.** Challenges in the wireless telesurgery use case.

| Challenge | Challenge Impact on the Use Case | Severity |
|---|---|---|
| SSLA | Data related to conducting remote operations or transmitting audio/video in good quality require adequate protection. Data require appropriate SSLA requirements. Strong security methods must be used to protect this data. | H |
| Contents to transfer | The selection of data for remote telesurgery of the patient is essential in the context of the patient's life. Moreover, data needed to establish a high-quality stream of people taking part in teleconferences are another aspect of continuing service. | H |
| Transfer initiation | Before the context transfer is started, the side that will initiate it must be selected: user equipment or a MEC instance. It is related to specifying the requirements when such a transfer could occur. Moreover, data transfers from the old MEC instance to the new one may be required. | H |
| Transfer process | The patient's data and telesurgery commands must be correctly migrated during the transfer process. It is essential because we can react quickly to rapid health changes. When we consider the audiovisual streaming migration process is needed to ensure continuous availability of service with high quality. In cases where specific data are missing, an attempt to obtain them should be made immediately. | H |
| Transfer management | Managing the transfer of telesurgery and a high-quality audiovisual is important for the proper operation of the service. Therefore, choosing which party is responsible for this process must be made. One of the challenges in this process will be the selection of the appropriate MEC instance where the data will be stored, and this instance will meet the requirements specified in the SSLA. | H |
| Isolation | The entire process of performing wireless telesurgery or streaming high-quality audiovisual images must be adequately secured—especially telesurgery. Thus, the isolation of such type during its transfer and storing and processing is required. | H |

### 6.5. Critical Communication (Emergency)

The use case is about communication in emergencies where a part of critical infrastructure can be destroyed or be out of order due to unpredicted issues or attacks. For various emergency forces in such circumstances, minimal communication services must be provided that are possible for a significantly large group of people in the exact location [76]. It is essential to reduce energy consumption on both mobile and network infrastructure sides because the regular power supply might be unavailable, and network devices might work on batteries or with a fuel-based power generator. In this service, a small RTT is demanded that would be eligible for voice communication or automation. The MEC infrastructure could be utilized to handover traffic between MEC cells or intercept traffic towards cloud services that could be handled by MEC service without external connection to the core network.

Due to the possible destruction of infrastructure, the security context should be governed by user applications and mobile devices of a particular purpose.

In an emergency, only a limited number of SSLA conditions are important, but they can be crucial for the emergency procedure (secrecy, service availability, authentication, and authorization) and post-incident procedures (data integrity, registration, and non-repudiation). Since MEC transferred data can be a source of some core information in post-emergency procedures (disaster recovery), all contents-related procedures are highly

severe. Isolation is necessary but not critical if other content-related security parameters are satisfied: confidentiality, integrity, authenticity, and service continuity.

In Table 10, there are security challenges for the use case presented for every area of the security context transfer considered in Section 5.

**Table 10.** Challenges in the critical communication (emergency) use case.

| Challenge | Challenge Impact on the Use Case | Severity |
|---|---|---|
| SSLA | The priority is service continuity, so SSLA is in shadow. However, after the emergency, some recovery procedures and analyses must be performed, so SSLA requirements should give such a possibility. | M |
| Contents to transfer | Well-defined content is the basis of proper service functioning. | H |
| Transfer initiation | Instant security context transfer could be the condition of the service functioning. | H |
| Transfer process | Correct security context transfer could be the condition of the service functioning. | H |
| Transfer management | Effective security context transfer management could be the condition of the service functioning. | H |
| Isolation | Isolation is not the priority of this service. | M |

### 6.6. City Surveillance

This use case aims to support city surveillance processes. Various devices such as CCTV cameras can collect data streams, IoT devices including drones or UAVs, or even personal cameras operated by patrolling robots, police, or other security officers [77]. Streams might contain, e.g., audio-video data, telemetry, measurements collected by motion sensors, and air quality information. Devices usually do not have sufficient computation power to perform advanced analytics, especially with near-real-time demand, so these tasks could be executed by MEC and services, enhancing the process with advanced data correlation techniques or AI/ML-based algorithms. Such a service should have HA and be isolated from other services due to the importance of the traffic. Surveillance processes are continuously executed daily, e.g., continuous water quality checks in smart sewage [77], and during special situations such as terrorist attacks, alarms, evacuations, and forest fires [78]. Each data stream analyzed in the MEC has its policy that allows or denies sharing this data stream with other stakeholders. The data might also be down-streamed to UE/robot.

High severity for 'contents to transfer' was picked because some AI or Machine Learning algorithms require data sets to work properly, including the history of the current data flow. These data must be available from the new MEC server, so a quick transfer is needed to maintain service continuity.

In Table 11, there are security challenges for the use case presented for every area of the security context transfer considered in Section 5.

**Table 11.** Challenges in the city surveillance use case.

| Challenge | Challenge Impact on the Use Case | Severity |
|---|---|---|
| SSLA | The content should be protected all the time; however, the needed security level is on average compared to, e.g., banking and healthcare. The constrained number of MEC services that should support these solutions is scoped by the city size and its closest neighborhood. Some devices, such as CCTV cameras, will use only a few MEC servers because their physical mobility is minimal, and eventual handovers might be a result of changes in the propagation channel itself. | M |
| Contents to transfer | Due to the scoped number of MEC servers, it is possible to maintain a common version of applications. Users' specific configuration and data should be migrated, related to the server-side network setup. In more advanced scenarios, where objects are tracked across many MEC servers, the information about the history is needed and should be shared with the current MEC server. | H |

**Table 11.** *Cont.*

| Challenge | Challenge Impact on the Use Case | Severity |
|---|---|---|
| Transfer initiation | The largest impact is on scenarios where historical information is needed, so the data context should be migrated (or shared between MEC servers). It can be mitigated by resource sharing and asynchronous data transfer in the first seconds after UE joins the new MEC server. | M |
| Transfer process | The connection can be reestablished, so the impact is low. Most data to be transferred are between MEC servers directly. The video stream captured by the device might be buffered during the migration time. | L |
| Transfer management | Due to the constrained number of MEC servers, it is easier to manage a shared policy for transfer management and to follow it, even with a simplified protocol that does not cover all flexibility and negotiation options. In the simplest scenarios, even no additional process is needed, or there is only a need to hand over the session created on the application layer. | M |
| Isolation | Medium impact. An average security level is needed. During a short unavailability period, data frames could be buffered on UE's side. | M |

*6.7. V2X Communication*

Vehicles in V2X scenarios can communicate with objects such as road infrastructure (V2I), other vehicles (V2V), pedestrian devices (V2P), and buildings. MEC-based service can be used as the trusted third party with MEC's computing power to collect and analyze aggregated data from various sources to produce knowledge about road traffic status; it can be simultaneously exposed to devices. This use case covers scenarios defined in [79], such as bird's eye view, vulnerable road user discovery, and cooperative collision avoidance; the last one is also considered in the literature as a mix of the cooperative vehicles and the inter-vehicle information exchange [80]. This use case generally requires high mobility with ultra-low response times, and decisions are crucial. The last part is challenging due to the untrusted environment, which includes unpredictable actors and situations—kids, terrorists, drunk persons, drivers with incorrect situation evaluation, and accidents. Vehicles' software might be updated in the air from the trusted repository in the MEC.

Due to the eventual consequences of accidents, including accidents with pedestrians, children, animals, other vehicles, etc., it is very important to have the service available on time when the device enters the new MEC cell. It is the reason for marking those challenges with high severity.

In Table 12, there are security challenges for the use case presented for every area of the security context transfer considered in Section 5.

**Table 12.** Challenges in the V2X communication use case.

| Challenge | Challenge Impact on the Use Case | Severity |
|---|---|---|
| SSLA | It depends on the security violation. Some security issues such as the lack of encryption might be temporarily accepted, and other issues such as MITM attacks detected between devices or between devices and MEC applications should be managed more rigorously. The information's authenticity and availability are crucial for this use case, and SSLA related to these functions must be satisfied to perform the service. Otherwise, the service should function in the safe mode. | H |
| Contents to transfer | In the new MEC server, there are probably other infrastructure devices that could be used in V2I scenarios and that should be prepared for communication with the vehicle; the same applies to pedestrians' personal devices. | H |
| Transfer initiation | Due to the relatively high speed of vehicles, it is crucial to perform the decision about contextual transfer as fast as possible. MEC servers should manage it because device does not have full knowledge about related infrastructure and which MEC server it is really connected to. Some not-crucial enhancements such as encryption and informative road signs might be enabled later in the switching process to minimize the switching time. | H |

**Table 12.** *Cont.*

| Challenge | Challenge Impact on the Use Case | Severity |
| --- | --- | --- |
| Transfer process | Vehicles should be prepared to work correctly without connecting to the MEC server or within a transition period between MEC servers. Fallback algorithms and mechanisms should be prepared at the application level. The MEC server should inform applications immediately when it knows that the context migration process cannot be completed in the assumed timeframe. | H |
| Transfer management | The actual MEC server could control the transfer process because it knows when it is needed to handover the vehicle to another MEC server (based on the actual history of the vehicle's location and the actual velocity of the vehicle); it also might have knowledge about which surrounded MEC server covers the place and where the vehicle is going. The target MEC server knows about surrounding objects, so it knows which relationships and communication channels will be used in the nearest future. | H |
| Isolation | Due to high availability needs, proper traffic isolation is strongly required. A lack of messages, big delays, or changes in messages might lead to traffic accidents. | H |

### 6.8. Automated Vehicles

This use case consists of Cooperative Awareness, Cooperative Sensing, and Cooperative Maneuvering scenarios [81] that can be used in the Logistics or Smart City verticals both on the road and off-road. It aims to provide vehicles that become autonomous and self-driving on the road. This use case is an extension of the V2X communication regarding the cooperation between vehicles and the ability to self-drive [82]. It also inherits many difficulties identified for the V2X use case, mostly related to high-quality requirements and an untrusted environment. In this scenario, it is even more critical to cooperate correctly with other devices and detect untrusted vehicles or dangerous road situations.

Due to the eventual consequences of accidents, including accidents with pedestrians, children, animals, other vehicles, etc., it is very important to have the service available on time when the device enters the new MEC cell. It is the reason for marking those challenges with high severity. Vehicles in this use case should be more self-sufficient than in the V2X use case. However, solving the computation complexity of problems requires support from edge or cloud systems.

In Table 13, there are security challenges for the use case presented for every area of the security context transfer considered in Section 5.

**Table 13.** Challenges in the automated vehicles use case.

| Challenge | Challenge Impact on the Use Case | Severity |
| --- | --- | --- |
| SSLA | It depends on the security violation. Some security issues such as the lack of encryption might be temporarily accepted, and other issues such as MITM attacks detected between devices or between device and MEC applications should be managed more strictly. The authenticity and availability of the information are crucial for this use case, and SSLA related to these functions must be satisfied to perform the service. Otherwise, the application running on the vehicle and the service itself must work in the safe mode because some minimal level functionalities of the service must be provided—e.g., gracefully stopping the vehicle on the road. | H |
| Contents to transfer | The scope includes data used in the V2X use case. It also contains the trust information about the vehicle that the MEC server obtained from the past, its own observations, and received from previous MEC servers. | H |
| Transfer initiation | Due to the relatively high speed of vehicles, it is crucial to perform the decision about context transfer as fast as possible. MEC servers should manage it because the device does not have full knowledge about related infrastructure and to which MEC server it is really connected. Some not-crucial enhancements such as encryption and informative road signs might be enabled later in the switching process to minimize the switching time. Direct communication between vehicles can be used for the switching time or while the MEC server is unavailable. | H |

| Challenge | Challenge Impact on the Use Case | Severity |
|---|---|---|
| Transfer process | Vehicles must be prepared to work correctly without connecting to the MEC server or within a transition period between MEC servers. Fallback algorithms and mechanisms must be prepared on the application level. MEC must inform applications and services immediately when it knows that the context migration process cannot be completed in the assumed timeframe. | H |
| Transfer management | The transfer process could be controlled by the actual MEC server because it knows when it is needed to handover the vehicle to other MEC servers (based on the actual history of the vehicle's location and the actual velocity of the vehicle); it also might have knowledge on which surrounded MEC server covers the place and where the vehicle is going. The target MEC server knows about surrounding objects knowing which relationships and communication channels will be used in the near future. Application layer data such as information about the trip, the car's current state, position, velocity, etc., also must be transferred to the target system. | H |
| Isolation | Due to extremely high availability and latency needs, proper traffic isolation is strongly required. A lack of messages, big delays, or changes in messages might lead to traffic accidents. | H |

*6.9. Importance and Difficulty of the Challenges*

In Table 14, we present a summary of the challenge's significance concerning the individual use cases discussed above. This significance is determined on a scale from 1 to 3, where "1" is very important, and "3" is not essential.

**Table 14.** The importance of the challenge.

| | SLA | Contents to transfer | Transfer Initiation | Transfer Process | Transfer Management | Isolation |
|---|---|---|---|---|---|---|
| Mobile-to-Bank (M2B) | 3 | 3 | 2 | 1/2 | 1/2 | 1 |
| Remote monitoring of health or wellness data through wireless devices | 1 | 1 | 2 | 1 | 2 | 1 |
| Wireless telesurgery | 1 | 1 | 1 | 1 | 1 | 1 |
| Critical communication (emergency) | 2 | 1 | 1 | 1 | 1 | 2 |
| City surveillance | 2 | 1 | 2 | 3 | 2 | 2 |
| V2X communication | 1 | 1 | 1 | 1 | 1 | 1 |
| Automated vehicles | 1 | 1 | 1 | 1 | 1 | 1 |

In Table 15, we present a table summarizing the difficulties of solving the challenge concerning the individual use cases discussed earlier. This difficulty is determined by a scale of 1 to 3, where "1" means "difficult" and "3" means "easy". Those two tables show that significant challenges are usually not easily solvable—except for the content transfer in e-health use cases or isolation in the M2B use case. Use cases based on the vehicle communications vertical (V2X, automated vehicles) have the highest number of complex and essential challenges. On the opposite side, the M2B (the BFSI vertical) and remote monitoring of health data (the e-health vertical) do not have challenges that are both difficult and important.

**Table 15.** The difficulty of solving the challenge.

| | SLA | Contents to Transfer | Transfer Initiation | Transfer Process | Transfer Management | Isolation |
|---|---|---|---|---|---|---|
| Mobile-to-Bank (M2B) | 3 | 3 | 1 | 3 | 3 | 3 |
| Remote monitoring of health or wellness data through wireless devices | 2 | 3 | 2 | 2 | 1 | 2 |
| Wireless telesurgery | 2 | 3 | 2 | 2 | 1 | 2 |
| Critical communication (emergency) | 2 | 2 | 2 | 1 | 1 | 1 |
| City surveillance | 3 | 2 | 2 | 3 | 3 | 2 |
| V2X communication | 2 | 2 | 1 | 1 | 2 | 1 |
| Automated vehicles | 2 | 2 | 1 | 1 | 2 | 1 |

## 7. Solutions Applicable to Resolve Chosen Challenges

In this section, we propose several solutions to resolve chosen challenges for the areas of interest of the security context transfer described in Section 6 for several use cases. The proposed potential remedies are motivated by use cases where the impact of the challenges is critical. As a summary of the section, we present some solutions for concrete use cases of mobile edge services.

### 7.1. SSLA

**Solution 1 (hardware incompatibility)** Formulate SSLA conditions in terms of performance and not resources. It could increase the spectrum of hardware solutions satisfying SSLA. From the architectural point of view, SSLA should be an abstraction layer between requirements and the solution with software or hardware implementation.

**Solution 2 (protecting PII)** Use containers to store and process PII. It can provide fixed protection levels and restricts PII data distribution, which helps satisfy GDPR constraints. This solution ensures a significant isolation level between container instances that might support multi-tenancy without mixing data between different application instances.

### 7.2. Contents to Transfer

**Solution 1. (data for proof of work, proof of quality)** Use blockchain solutions to protect such data [83,84]. It is a type of distributed data repository providing data integrity. It enables access to some past data even if it is not instantly transferred between MEC instances.

**Solution 2. (security credentials)** Use centralized security solutions (KDC, PKI, and OAuth 2.0) or dedicated MEC-related solutions, e.g., MEC Enabler [85]. It can help decrease the volume of data transfer and increase and unify the level of security of all MEC instances, which solves some SSLA conditions. With specific extensions, such a solution also guarantees a reduction in the latency in the provision of network services; see [86].

### 7.3. Transfer Initiation

**Solution 1** Mathematical model of transfer time optimization taking into account:

- User device location;
- User movement history/user path up to present time;
- MEC servers distribution related to possible methods;
- User characteristics (expected future time to use the service, etc.);
- MEC servers present workload;
- Etc.

**Solution 2** Use some AI method to decide about transfer initiation. The decision system would be trained using patterns, including the past behavior of users and past states of the workload of the MEC servers.

**Solution 3** Define a policy specifying the conditions to start the context transfer initiation. An external entity should probably handle this.

**Solution 4** The actual MEC server triggers the migration because it has existing knowledge about UEs position, signal strength, and current capacity and might obtain information about the capacity of other MEC servers. UE could indicate that the current QoE decreased and can suggest migrating to another MEC.

**Solution 5** Perform an agile approach. Services could be designed so that they can almost always migrate to other MEC servers, and after the migration, features are enabled if SSLA is satisfied. This approach enables fast migration and supports asynchronous enabling features to provide a service with improved service offerings.

*7.4. Transfer Process*

**Solution 1** Design as ready for faults. Errors in the context of transmission will occur and should be managed. The process should support retries of operations, and applications should support the safe mode or decide not to provide a service to the client if the SSLA is not established with the required level.

**Solution 2** The security context should be stored longer for complex scenarios if one cannot repeat the context creation process. The time could be shortened in a case of a lack of resources due to reasons with respect to the law (e.g., when GDPR data are stored and no longer used) or when the client's demands are to clean up the data.

**Solution 3** Updating existing contexts. When UE arrives again and its context still exists in the MEC server, it should be reused if it makes sense. Implementation details are important here, e.g., the session keys should be changed, but firewall rules might remain untouched. Data sets might be re-transferred or updated—it depends on the internal architecture of the dataset scale of applied changes.

**Solution 4** Use asynchronous migration to reduce the time needed to perform a service. Even with reduced scope and functionalities, the migration should be performed asynchronously.

**Solution 5** Use remote access to the data. While needed data are still under the synchronization (migration) process, the application could use the data stored in the previous MEC server.

*7.5. Transfer Management*

**Solution 1** Use a hysteresis loop to avoid unnecessary migrations. With this solution, the user will not be transferred to the new MEC instance until it is really necessary or worthy from the MEC instance owner's perspective.

**Solution 2** MEC server selection should be context-aware. In V2X scenarios, the vehicle should be migrated to the same MEC with the infrastructure that the vehicle can meet on the road. The most important factor in the city surveillance use case is the current capacity, especially the computation power on the MEC side. The MEC server might present a set of available metrics, and the application could decide their importance level.

**Solution 3** Follow an application's policy for migration. The application (service) knows best if migration to an insecure and unsafe place is a show-stopper. In some circumstances, not providing a service is the best option. It is reasonable to provide a service with a degraded security level or without a full security setup applied in other scenarios.

**Solution 4** Rely on abstractions and interfaces. This approach enables the possibility of using different implementations of the same security function to obtain the same or similar effect. For instance, it is possible to exchange encryption algorithms. For data with small to medium expected use time, it is also possible to use reduced key length—e.g., AES with 128 bits instead of 256 bits for data that will be stored for a couple of months. In such

a situation, the problem of managing implementation differences that MNVO might not fully know relies on suppliers and service providers.

*7.6. Isolation*

**Solution 1** Use containers with appropriate protection (sandboxing, firewalling, and homomorphic encryption). The transfer of encrypted containers. Tunneling in communication.

**Solution 2** Select proper cloud solutions that provide methods for data protection. Data might be processed on separate virtual machines, containers, databases, in different network segments, transferred over VPN, etc. Those solutions could support data protection at rest, during transmission, and during use.

**Solution 3** Ensure user access controls with authentication and identity separation. Each user or technical integration should use a separate identity such as a different account, certificate, token, etc.

*7.7. Examples of Solutions Applicable to the Challenges in Use-Cases*

This subsection covers descriptions of selected solutions for use cases and challenges defined earlier in this paper.

7.7.1. Isolation Possibility in Azure Cloud Computing

Microsoft Azure is a hyper-scale public multi-tenant cloud services platform that provides access to a feature-rich environment. It incorporates the latest cloud innovations such as artificial intelligence, machine learning, IoT services, big-data analytics, intelligent edge, and much more to help increase efficiency and unlock insights into operations and performance. A multi-tenant cloud platform implies that multiple customer applications and data are stored on the same physical hardware. Azure uses logical isolation to segregate your applications and data from other customers. This approach provides multi-tenant cloud services' scale and economic benefits while rigorously helping prevent other customers from accessing data or applications. Azure addresses the perceived risk of resource sharing by providing a trustworthy foundation for assuring multi-tenant, cryptographically specific, logically isolated cloud services using a standard set of principles:

1. User access controls with authentication and identity separation;
2. Compute isolation for processing;
3. Networking isolation, including data encryption in transit;
4. Storage isolation with data encryption at rest;
5. Security assurance processes embedded in service designs to correctly develop logically isolated services.

Multi-tenancy in the public cloud improves efficiency by multiplexing resources among disparate customers at low costs; however, this approach introduces the perceived risk of resource sharing. Azure addresses this risk by providing a trustworthy foundation for isolated cloud services using a multi-layered approach offering methods available in cloud computing to protect data at rest.

7.7.2. Application Repositories for Application Migration

In some situations, installing a MEC application that was not available previously on the MEC server might be needed. From the service provider's perspective, it is costly and complex to test the application in every environment and to set it up separately. To solve this problem, container-based solutions might be used to provide well-defined software (from the MEC operator perspective). On the other hand, the MEC operator might expose an example VM image that is a reference environment for service providers to make their applications, where containers might be deployed—in a direct way, e.g., with Docker, or with some clustering based on Kubernetes, or any other solution. This abstraction layer will help prepare stable solutions, simplify system integration, and reduce the needed time to market.

### 7.7.3. Security Context Transfer Managed by User Application in BFSI Mobile Transactions

Small value transactions in mobile banking are performed in a contactless manner, with only interactions of the User Device (application) and Point of Sale. It increases the payment system's performance but causes new security risks (device cloning, exceeding payment limits, masquerade, etc.). After the relocation of the end user, his/her service should be transferred to another MEC location (instance), which could also help in operations that are more critical than small payments (e.g., access to the user's bank account and ordering operations). The hardware components of the user's devices can be used to increase the security of the context transfer (secure processors SGX, secure memory, or secure units in mobile devices, etc.). Such solutions enable building the chain of trust between the User Device and a BFSI institution to authorize all operations governed by the user's device. It may also securely authorize the security context transfer between MEC instances initiated by the user's device.

## 8. Summary and Future Work

In the paper, we dealt with the problem of migrating services between edge servers and guaranteeing the level of security in this process. We presented the cloud computing ecosystem in which the service migration takes place along with the basic concept of guaranteeing the level of such a service, which is the Service Level Agreement, and in the case of the security level, the Security Service Level Agreement. Aiming to develop a reliable and widely applicable security context transfer procedure, we reviewed the latest work in the field of service transfer in the edge cloud environment, particularly using MEC technology, as well as the latest recommendations of standardization organizations in this area. As a result, we proposed a security context migration procedure that is independent of security technologies used and that is easy to apply in different instances of edge servers. Finally, we approached the issue of the transfer of the security context and the challenges related to its practical implementation. We decided to group the process of transferring the security context into six areas of interest, in which we presented challenges related to its feasibility and security in more detail.

To verify our considerations more practically, we analyzed seven specific verticals of 5G MEC mobile networks and mapped the proposed areas of interest of the security context transfer to selected specific use cases trying to assign the formulated challenges. We also tried to indicate the possibilities of solving the identified security challenges using known secure computing technologies in cloud computing and mobile networks.

The literature studies conducted in this article and our related research are preliminary, but at this stage, they show the possibilities of further work leading to specific security solutions and responding to the identified challenges. It may involve a deeper analysis of the feasibility of the proposed security solutions and an attempt to solve selected challenges in specific use cases fully. The reliable and effective migration of services between different instances of edge servers requires the preparation of routine security procedures that are useful in a heterogeneous cloud environment. Work on this issue, carried out by standardization organizations, is already well-advanced, as we showed in our review of state-of-art methods. However, the subject of future research and the search for new solutions remains as an extensive range of issues related to facilitating standard schemes and finding lightweight, high-speed solutions and schemes independent of the complete network management infrastructure. Examples of such problems that require solutions are listed as follows:

- How do we build a security context in an agile way that is easy to reuse later or could be used as a template for new user instances? The new user has an empty history, so his reputation cannot be established, limiting the security context data. Likewise, an application's operation interruption decreases the recorded data's usability.
- How do we expose a security context offering for the MEC application/service/new user in terms of choosing the appropriate security level for available service offerings?

- How can a negotiated SSLA framework be adapted for specific services with different security context values?
- Is it possible to integrate security context migration with the Single Sign On feature? How do we reconcile security context migrations with the requirements of user privacy?
- How can the system be protected from malicious users who force frequent security context migrations? How do we avoid blurring the user's history in such a situation?

**Author Contributions:** Conceptualization, K.B. and J.-P.W.; funding acquisition, K.B. and J.-P.W.; validation, W.N., R.A., K.B., and J.-P.W.; investigation, T.W.N., M.S., and Z.K.; writing—original draft preparation, T.W.N., M.S., and Z.K.; writing—review and editing, W.N., T.W.N., M.S., Z.K., R.A., K.B., and J.-P.W.; supervision, Z.K., K.B., and J.-P.W.; project administration, Z.K. and K.B. All authors have read and agreed to the published version of the manuscript.

**Funding:** The research leading to these results received funding from the European Union's Horizon 2020 research and innovation programme under grant agreement no.871808 (5G PPP project INSPIRE-5Gplus). The paper reflects only the authors' views. The Commission is not responsible for any use that may be made of the information it contains.

**Data Availability Statement:** Not applicable.

**Conflicts of Interest:** The authors declare no conflict of interest.

## Abbreviations

The following abbreviations are used in this manuscript:

| | |
|---|---|
| 2G | Second-generation cellular network; |
| 5G | Fifth-Generation mobile network; |
| AAA | Authentication, Authorization, and Accounting; |
| ABAC | Attribute-Based Access Control; |
| AES | Advanced Encryption Standard; |
| AI/ML | Artificial Intelligence or Machine Learning; |
| AMS | Application Mobility Service; |
| API | Application Programming Interface; |
| ASA | Average Speed to Answer; |
| BFSI | Banking, Financial Services, and Insurance; |
| CCTV | Closed Circuit Television; |
| CIA | Confidentiality, Integrity, and Availability; |
| CSC | Cloud Service Customer; |
| CSP | Cloud Service Provider; |
| DAC | Discretionary Access Control; |
| DBF | Data Base File; |
| DLP | Data Loss Prevention; |
| eMBB | enhanced Mobile Broadband; |
| ETSI | European Telecommunications Standards Institute; |
| FCR | First-Call Resolution; |
| GDPR | General Data Protection Regulation; |
| HA | High Availability; |
| IEC | International Electrotechnical Commission; |
| IP | Internet Protocol; |
| ISO | International Organization for Standardization; |
| ITS | Intelligent Transport System; |
| KDC | Key Distribution Center; |
| KPI | Key Performance Indicator; |
| M2B | Mobile to Bank; |
| MAC | Mandatory Access Control; |
| MEC | Multi-access Edge Computing; |
| MNO | Mobile Network Operator; |
| MTBF | Mean Time Between Failures; |

| | |
|---|---|
| MTTA | Mean Time to Acknowledge; |
| MTTC | Mean Time to Contain; |
| MTTI | Mean Time to Investigate; |
| MTTD | Mean Time to Detect; |
| MTTR | Mean Time To Recover; |
| MTTR | Mean Time to Resolve; |
| MVNO | Mobile Virtual Network Operator; |
| NIST | National Institute of Standards and Technology; |
| OEM | Original Equipment Manufacturer; |
| PII | Personally Identifiable Information; |
| PIN | Personal Identification Number; |
| PKI | Public Key Infrastructure; |
| QoS | Quality of Service; |
| RAID | Redundant Array of Independent Disks; |
| RBAC | Role-Based Access Control; |
| SAMOD | Shared Automated Mobility On-Demand; |
| SIEM | Security Information and Event Management; |
| SLA | Service Level Agreement; |
| SSLA | Security Service Level Agreement; |
| SLO | Service Level Objectives; |
| SQO | Service Qualitative Objectives; |
| TAT | Turn-Around Time; |
| TEE | Trusted Execution Environment; |
| TLS | Transport Layer Security; |
| TRT | Total Resolution Time; |
| TSF | Time Service Factor; |
| TTQ | Alarm Time to Qualify; |
| TTT | Alarm Time to Triage; |
| UAV | Unmanned Aerial Vehicle; |
| UE | User Equipment; |
| URLLC | Ultra Reliable Low Latency Communications; |
| V2I | Vehicle to Infrastructure; |
| V2P | Vehicle to Pedestrian; |
| V2V | Vehicle to Vehicle; |
| V2X | Vehicle to everything; |
| VIS | V2X Information Service; |
| VLAN | Virtual Local Area Network; |
| VM | Virtual Machine. |

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
