# Peer review of "Security Context Migration in MEC: Challenges and Use Cases"

_electronics, doi:10.3390/electronics11213512_

Round 1

Reviewer 1 Report

In the paper, the authors have studied the concept of Service Level Agreement for clouds and mobile computing and transferring the Security Context. They have got to the issue of the transfer of the Security Context and the challenges related to its practical implementation. Also, they have decided to group the challenges of transferring the Security Context into six categories, in which they have presented in more detail the challenges related to the feasibility and security of this process. Further, they have considered nine specific use cases of 5G Multi-Access Edge Computing mobile networks and mapped these categories to selected specific use cases. Moreover, they have tried to indicate the possibilities of solving the identified security challenges with the use of known technologies of secure computing in cloud computing and mobile networks. The reviewer finds the submitted manuscript adds to the existing body of knowledge and it is interesting to read after considering the following comments:

Point 1: Some of the paragraphs need to be proofread so that the grammatical errors can be corrected.

Point 2: Rewriting the abstract is recommended. 

Point 3: Section 3 'Related work on context migration' need to be condensed and merged with Section 1 'Introduction'.

Point 4: It is suggested to add some relevant references, such as doi: 10.1109/EExPolytech.2019.8906885 and 10.1109/WoWMoM54355.2022.00035

Point 5: Tables 15 and 16 in Section 9 better to move it to the end of section 8 with some description.

Author Response

Thank you for your opinion and remarks about the proposed manuscript. Below is our reply to your suggestions.

Point 1: We have corrected the spelling and grammar of the paper.

Point 2: The abstract has been significantly changed.

Point 3: In the new version of the paper, we have expanded the literature with fresh articles suggested by reviewers and new standards on MEC security recently published by ETSI. In this situation, condensing the description of the state-of-the-art and related works is too hard. We've only improved the presentation in this SOA, highlighting the work of other authors on migrating applications between different instances of edge servers. The Introduction has also been modified to underline the paper’s novelty.

Point 4: Proposed references were added and cited.

Point 5: Tables were moved to section 6, and an additional discussion of those tables was added. We decided to move them to this section because section 6 is the first section that covers the challenges that are mentioned in those tables. Sections 7 & 8 have been merged into section 7.

Reviewer 2 Report

In this paper, the authors present the state-of-the-art research on the migration of the security context between service instances in Edge/MEC servers, specify steps of the migration, procedure, and identify new security challenges inspired by use cases of 5G virtual industries.

Well-written article and the presentation of the work is good. However, the following are a few concerns, which are required to be considered in the revision to further improve the quality of the manuscript.

Comments:

1.    Some more recent and relevant papers regarding the security issue via emerging technologies like blockchain technology could be cited to support the literature review part of the paper.

a.       https://doi.org/10.1016/j.seta.2022.102248

b.    https://dl.acm.org/doi/fullHtml/10.1145/3474552

c.       https://ieeexplore.ieee.org/document/9811428/

d.       https://doi.org/10.1109/ACCESS.2020.3028240

2.    The manuscript is found to have write-up issues.  Thus, requires thorough proofreading.

3.    Further clarification on future work is needed.

Author Response

Thank you for your opinion and remarks about the proposed manuscript. Below is our reply to your suggestions.

Point 1: Proposed references were added and cited.

Point 2: We have corrected the spelling and grammar of the paper.

Point 3: The summary and future work chapter have been modified.

Reviewer 3 Report

This paper aims to present the state of research on the migration of the Security Context between service instances in Edge/MEC servers, specify steps of the migration procedure, and identify new security challenges inspired by use cases of 5G virtual industries. However, there are some problems in this paper, as follows:

1.    The abstract has the problem of inaccurate expression (e.g., the third-person pronoun) and improper generalization of the main research content of the article.

2.    Many of the tables in this paper are poorly formatted for viewing, for example, Table 15 and so on.

3.    In section 7, the description of some solutions is too simplistic and does not reflect the specificity, and there are similar problems in section 8.

4.    Some grammar mistakes, for example, “These kind of services works on daily basis” in the introduction and so on.

Author Response

Thank you for your opinion and remarks about the proposed manuscript. Below is our reply to your suggestions.

Point 1: The abstract has been significantly changed.

Point 2: Tables have been rearranged with horizontal text and a bigger margin between text and table lines to increase their readability. 

Point 3: Sections 7 and 8 were merged into a single chapter since those chapters cover very similar topics. The solutions’ descriptions have been extended.

Point 4: We have corrected the spelling and grammar of the paper.

Reviewer 4 Report

The authors have presented their work in good detail however, this paper is more like a report than a research article. The novelty of the work is not clear. Authors need to clearly mention how this work is novel or new and how it is compared with the similar works in the literature. The structure of the paper is more like a report and thus it lacks reader's interest. 

Author Response

Thank you for your opinion and remarks about the proposed manuscript. 

A paragraph about the novelty of the work and a comparison with other approaches has been added in the Introduction.

Round 2

Reviewer 2 Report

Well done! No further comments